# Genetic polymorphisms in the serotonin, dopamine and opioid pathways influence social attention in rhesus macaques (*Macaca mulatta*)

**Emmeline R. I. Howarth**[1,2]*, **Isabelle D. Szott**[1], **Claire L. Witham**[3], **Craig S. Wilding**[4], **Emily J. Bethell**[1]*

1 Research Centre in Brain and Behaviour, School of Biological and Environmental Sciences, Liverpool John Moores University, Liverpool, United Kingdom, 2 Department of Biological Sciences, University of Chester, Chester, United Kingdom, 3 Centre for Macaques, Harwell Institute, Medical Research Council, Salisbury, United Kingdom, 4 Biodiversity and Conservation Group, School of Biological and Environmental Sciences, Liverpool John Moores University, Liverpool, United Kingdom

* e.howarth@chester.ac.uk (ERIH); e.j.bethell@ljmu.ac.uk (EJB)

**Data Availability Statement:** All relevant data are within the paper and its Supporting information files.

## Abstract

Behaviour has a significant heritable component; however, unpicking the variants of interest in the neural circuits and molecular pathways that underpin these has proven difficult. Here, we present a comprehensive analysis of the relationship between known and new candidate genes from identified pathways and key behaviours for survival in 109 adult rhesus macaques (*Macaca mulatta*). Eight genes involved in emotion were analysed for variation at a total of nine loci. Genetic data were then correlated with cognitive and observational measures of behaviour associated with wellbeing and survival using MCMC-based Bayesian GLMM in R, to account for relatedness within the macaque population. For four loci the variants genotyped were length polymorphisms (*SLC6A4* 5-hydroxytryptamine transporter length-polymorphic repeat (*5-HTTLPR*), *SLC6A4 STin* polymorphism, *Tryptophan 5-hydroxylase 2* (*TPH2*) and *Monoamine oxidase A* (*MAOA*)) whilst for the other five (5-hydroxytryptamine receptor 2A (*HTR2A*), *Dopamine Receptor D4* (*DRD4*), *Oxytocin receptor* (*OXTR*), *Arginine vasopressin receptor 1A* (*AVPR1a*), *Opioid receptor mu(μ) 1* (*OPRM1*)) SNPs were analysed. *STin* genotype, *DRD4* haplotype and *OXTR* haplotype were significantly associated with the cognitive and observational measures of behaviour associated with wellbeing and survival. Genotype for *5-HTTLPR*, *STin* and *AVPR1a*, and haplotype for *HTR2A*, *DRD4* and *OXTR* were significantly associated with the duration of behaviours including fear and anxiety. Understanding the biological underpinnings of individual variation in negative emotion (e.g., fear and anxiety), together with their impact on social behaviour (e.g., social attention including vigilance for threat) has application for managing primate populations in the wild and captivity, as well as potential translational application for understanding of the genetic basis of emotions in humans.

**Funding:** This research was supported by a National Centre for the Replacement, Refinement and Reduction of Animals in Research (NC3Rs) grant NC/L000539/1to EJB. ERIH was supported by a Liverpool John Moores PhD studentship. The funders had no role in study design, data collection and analysis, decision to publish, or preparation of the manuscript.

**Competing interests:** The authors have declared that no competing interests exist.

## Introduction

In humans (*Homo sapiens*) and many other animal species, a significant heritable component of personality [1, 2] and behaviour [3, 4] has been demonstrated. Whilst identification of genetic variants that are consistently associated with such traits would allow a better understanding of the specific neural circuits and molecular pathways that underpin emotional behaviours [5, 6], due to the complexity and number of these pathways involved in behavioural responses, unpicking the variants of interest has often proven difficult.

Phenotypic traits, such as personality, are both quantitative (non-binary) and usually polygenic [7]. Identifying the loci underpinning behavioural traits has sometimes involved their study in mapping populations created by experimental crosses [8]. Such quantitative trait locus (QTL) analysis has led to the identification of specific QTL relating to, for example. avoidance behaviours [9], anxiety [10] and aggressive behaviour [11]. However, QTL are typically broad, and require substantive follow-up work to identify causal genes within the identified QTL. More recently, genome wide association studies (GWAS) in which associations between genotype and phenotype are examined in populations of unrelated individuals have been utilised to study a range of behavioural conditions (e.g., risk-taking [12], antisocial behaviour [13], anxiety [14]).

Whilst there are numerous genomic resources for rhesus macaques (*Macaca mulatta*) [15] due to the genomic nature of the data needed, QTL and particularly GWAS can, however, be expensive. Candidate gene studies involving genotyping variants at a smaller number of preselected candidate loci have long been a more affordable and accessible approach for genetic analysis [16].

With this cost in mind, this study aimed to provide the most comprehensive analysis to date of the relationship between individual candidate genes from identified pathways and associated behaviours and vigilance for threat in adult rhesus macaques. We first review the primate literature for each pathway (with relevant human studies also included, especially where primate data are lacking) to justify our choices of candidate loci. Table 1 provides a summary of the pathways, genes, and variants of interest in this study and an overview of whether published primate and human data are available.

### Serotonin pathway

A key candidate gene in the study of anxiety and stress-related disorders is the serotonin transporter (5-hydroxytryptamine transporter or *5-HTT*) gene (also known as *SLC6A4* (Solute Carrier Family 6 Member 4)). Serotonin (5-hydroxytryptamine or 5-HT) is a neurotransmitter and 5-HTT regulates the signaling and concentration of synaptic 5-HT [18]. The neural circuitry controlling temperament and mood relies on 5-HT synapses and disturbances in this system result in many psychiatric disorders in humans [18]. *SLC6A4* displays three well-studied polymorphisms in humans: a guanine/thymine (G/T) SNP in a non-coding 3'-untranslated region (3'-UTR), an insertion / deletion polymorphism in the promoter region (the 5-HTT length-polymorphic repeat or *5-HTTLPR*) and a 16–17 bp variable number of tandem repeats (VNTR) polymorphism located in intron 2 (*STin*) [19, 20].

The most widely studied polymorphism is the *5-HTTLPR* for which, in humans, the predominant alleles are a 14-repeat short allele (s-allele) and a 16-repeat long allele (l-allele) [18, 21]. The s- and l-alleles differ in their rate of serotonin transporter transcription, with the s-allele having a lower rate of transcriptional efficiency than the l-allele, such that individuals homozygous for the s-allele or who are heterozygous have around a 65% lower 5-HTT mRNA expression level than homozygous l-allele individuals [21, 22]. A similar length variable allelic system is found in rhesus macaques, though here, the alleles are a 23-repeat s-allele and a

**Table 1. Candidate SNP or length variants for behavioural phenotypes in primates.** Further details are provided in the text. Variant ID refers to the location in the *Macaca mulatta* genome (Mmul_10 assembly [17]).

| Pathway | *Gene* | Variant ID (SNP only) / location (LP only) | Type of variant | Previous published studies |
|---|---|---|---|---|
| **Serotonin** | *SLC6A4* 5-hydroxytryptamine transporter length-polymorphic repeat (*5-HTTLPR*) | 16:25897958–25898456 | Insertion / deletion LP | Primates: ✓ Humans: ✓ |
| | *SLC6A4 STin* polymorphism | 16:25,882,595–25,882,709 | 16–17 bp VNTR LP | Primates: X Humans: ✓ |
| | *Tryptophan 5-hydroxylase 2 (TPH2)* | 11:71,601,619 | 159 bp insertion LP (TPH2IP) in the 3'-UTR | Primates: ✓ Humans: ✓ |
| | 5-hydroxytryptamine receptor 2A (*HTR2A*) | rs80365915 | Missense SNP | Primates: X Humans: X |
| | | rs80363349 | 5' UTR SNP | |
| | | rs196407124 | 5' UTR SNP | |
| **Dopamine / serotonin** | *Monoamine oxidase A (MAOA)* | X:43,576,386–43,576,511 | Length | Primates: ✓ Humans: ✓ |
| **Dopamine** | *Dopamine Receptor D4 (DRD4)* | rs300413141 | Upstream SNPs | Primates: X Humans: X |
| | | rs1079355788 | | |
| | | rs290724315 | | |
| | | rs301203363 | | |
| **Oxytocin** | *Oxytocin receptor (OXTR)* | rs196783445 | Intronic SNPs | Primates: X Humans: X |
| | | rs292502465 | | |
| | | rs300857875 | | |
| | | rs308701533 | | |
| | | rs292035217 | | |
| | | rs302789768 | | |
| | | rs283226059 | | |
| | *Arginine vasopressin receptor 1A (AVPR1a)* | 11:62,913,090 | 8bp repeat | Primates: X Humans: X |
| **Opioid** | *Opioid receptor mu(μ) 1 (OPRM1)* | rs195917455 | | Primates: ✓ Humans: ✓ |

Abbreviations: **SNP**–single-nucleotide polymorphism; **LP**- length polymorphism; **UTR**–untranslated region; **VNTR**—variable number of tandem repeats.

24-repeat l-allele. Whilst the effect of these alleles on transcriptional efficiency *in vitro* has not been studied, since 5-HTT genotype plays a role in serotonin regulation, which affects hypothalamic-pituitary-adrenal (HPA) axis function [23] phenotypic effects of genotype at this locus are expected and suggested in some studies. S-allele homozygous infant and juvenile monkeys (n = 128) displayed more anxious and threatened behaviour than l-allele carriers during novel fruit, human intruder, remote-controlled car, and free play tests [24]. Also, interactions between genotype and thalamic levels of 5-HTT, and dysregulation of both the HPA axis and serotonergic system, have been shown to be associated with negative mood states [25], the control of arousal, depression, and anxiety [26, 27]. In a study testing the effect of chronic stress caused by social conflict on 29 female rhesus macaques, *5-HTTLPR* genotype was found to influence cortisol response [28]. Individuals homozygous for the s-allele had higher hair cortisol compared to heterozygous or l-allele homozygous individuals. As chronic stress and heightened cortisol levels are associated with the onset of depression in adult female macaques [29, 30], they may be more susceptible to depression if they have low serotonin transporter efficiency (s-allele) and a history of stress.

By contrast to *5-HTTLPR*, the function of the VNTR polymorphism *STin* is less well studied [31]. In humans and primates, including macaques, there are several known alleles for *STin* [32, 33]. In humans, the three most common are *STin*2.9, *STin*2.10 and *STin*2.12, containing

9, 10 and 12 copies of the VNTR element respectively [31, 33]. *In vitro* studies with human embryonic stem cells have shown genotype-dependent reporter gene expression in embryonic stem cells with *STin*2.12 increasing gene expression 29-fold compared to *STin*2.10 [34]. Phenotypic studies have explored the association of these polymorphisms with e.g., tobacco use [20] and migraine susceptibility [33]. The *STin*2.12 allele has also been associated with obsessive–compulsive disorder (OCD) [35]. However, there have been discrepancies and conflicting data published on the effect of the *STin* alleles on psychological disorders [36, 37]. For example, Kaiser et al [38] reported a six-fold increase in the risk of developing a subtype of schizophrenia in human patients that carry the *STin*2.9 allele, with the allele being significantly associated with an increased risk of unipolar disorder and depression [39], while the *STin*2.10 allele has been reported as a predictor of suicide attempts in female members of the Dubla tribe of Daman [40].

The *tryptophan 5-hydroxylase 2* gene (*TPH2*) is also within the serotonin pathway [41]. *TPH2* encodes the protein catalyst, tryptophan hydroxylase, for the first and rate-limiting step in the biosynthesis of serotonin [42]. In macaques there are several known *TPH2* polymorphisms: two mononucleotide repeats, one dinucleotide repeat, 17 SNPs and, a 159 bp insertion polymorphism (*TPH2IP*) in the 3'-UTR [43]. In luciferase reporter gene assays the *TPH2IP* had a profound effect on gene expression, with the l-allele driving substantially higher gene expression than the s-allele. In humans, similar 3'-UTR length polymorphisms have been associated with ADHD [44], bipolar disorder [45] and suicide [46] and, in rhesus macaques, with altered HPA axis functioning and aggressive behaviour [47].

The 5-*hydroxytryptamine receptor 2A* gene (*HTR2A*) encodes the 5-HT$_{2A}$ receptor for serotonin, which is expressed mainly in the brain, including in the hippocampus, olfactory tubercle, nucleus accumbens, caudate nucleus and neocortex [48]. Polymorphisms in *HTR2A* have been associated with neuropsychiatric disorders including impulsive behaviour and schizophrenia [49, 50]. For example, in humans, carriers of the minor A-allele of a SNP $\approx$2kb upstream of *HTR2A* experienced greater depressive symptoms than individuals who are homozygous for the G-allele [51].

No specific candidate polymorphisms have been previously identified in macaques. Here, sequencing of the 5'UTR and first coding exon was undertaken to genotype three SNPs, one exonic (rs80365915) and two SNPs rs80363349 (A/G) and rs196407124 (A/C) in the 5'-UTR. They have not been previously evaluated for association with anxiety or stress-related disorders.

## Multiple pathways (serotonin & dopamine)

Monoamine oxidase A is involved in the degradation of circulating serotonin [52]. This mitochondrial enzyme catalyses the oxidative deamination of amines, including serotonin, dopamine, noradrenaline, and adrenaline and is encoded by the *monoamine oxidase A* gene (*MAOA*) [53]. A key polymorphism associated with aggressive behaviour in rhesus macaques are 5-, 6- and 7–18 bp repeat alleles located within the transcriptional control region of the *MAOA* gene [54]. *MAOA* is an X-linked gene and females can be heterozygous or homozygous while males can only be hemizygous (only one member of the chromosome pair or segment is present rather than two) [53]. The 5- and 6- repeat heterozygous, homozygous, or hemizygous variants have a significantly higher activity than the 7-repeat, resulting in a lower rate of degradation of circulating serotonin in 7-homozygotes [54, 55]. There is little *in vitro* or *in vivo* evidence of whether 5/7 and 6/7 repeat heterozygous variants have high or low activity, therefore, it is not known whether they result in a higher or lower rate of serotonin degradation [54, 55]. In the present study, the 5-, 6- and 7–18 bp repeat alleles have been investigated for their link to anxiety and attention bias (AB).

## Dopamine pathway

Dopamine (3, 4-Dihydroxytyramine) is associated with reward-motivated behaviour and memory [56]. Dopamine is a catecholamine that can act as a hormone which is released from the hypothalamus, or a neurotransmitter that activates the dopamine receptors [41, 57, 58]. There are five known dopamine receptors (Dr1-5) subdivided into two categories: D1-like receptors (Dr1 and Dr5) and D2-like receptors (Dr2-4). The *Dopamine Receptor D4* gene (*DRD4*) encodes the G-protein coupled D4 subtype of the dopamine receptor [59]. The receptor activates pertussis toxin-sensitive G-proteins [60], inhibits adenylyl cyclase activity and mediates dopamine activity in the central nervous system (CNS) [61]. In humans, *DRD4* contains several polymorphisms including a 48 bp VNTR in the third exon of the gene with the 7-repeat allele associated with psychiatric disorders including ADHD, bulimia, and alcoholism [62, 63]. This allele is also associated with anger, aggression, and delinquency in humans [64, 65] and avoidance of mothers and conspecifics in juvenile rhesus macaques [66]. Here, we sequenced a region of the *DRD4* promoter region to look for associations between alleles at four SNPs (rs30041314, rs1079355788, rs290724315, rs301203363) and AB.

## Oxytocin pathway

Oxytocin is a neuromodulatory posterior pituitary hormone that is associated with increased parasympathetic functioning and plays a counter-regulatory role in stress and fear responses [67, 68]. The *oxytocin receptor* (*OXTR*) is a potential key candidate gene in the study of oxytocin levels and the downstream effects of these. *OXTR* encodes G-protein coupled receptor proteins for oxytocin. Polymorphisms in human *OXTR*, including the most studied *OXTR* SNPs rs53576 and rs2254298, are associated with social behaviour and emotional responsiveness [69], reduced positivity [70], and poor social recognition skills [71]. Both rs53576 and rs2254298 are intron variants. For rhesus macaques, there are no previously characterised SNPs for *OXTR*.

*OXTR* encodes an oxytocin receptor, a member of a subclass of peptide receptors that also includes the arginine vasopressin receptors (1A, 1B, 2). Arginine vasopressin is a neuropeptide and the *arginine vasopressin receptor 1A* gene (*AVPR1a*) encodes the vasopressin V1a receptor. There are three known length variant polymorphisms in humans: RS1 a $(GATA)_{14}$ tetranucleotide repeat, RS3 a complex $(CT)_4$-TT-$(CT)_8$-$(GT)_{24}$ repeat and STR1 a $(GT)_{25}$ dinucleotide repeat [72]. RS1 and RS3 are located within two blocks, known as DupA and DupB respectively, of a ~350 bp tandem duplicate region [73]. Only the DupB region, and therefore RS3, is found in rhesus macaques [74]. Variation within RS3 is associated with altruism [75], pair-bonding behaviour [76], social behaviour including sibling conflict [77] and reproductive behaviour [78]. Walum et al [79] reported 11 RS3 alleles in humans (320, 330, 332, 334, 336, 338, 340, 342, 344, 346, 348 bp). The 334bp allele was the most common allele in the study (40%) and was associated with marital problems with homozygous 334 individuals having double the risk of a marital crisis compared to 334 non-carriers. In this study, *AVPR1a* RS3 alleles were investigated for their association with anxiety and stress-related behaviours.

## Opioid pathway

The endogenous opioid system, including the opioid receptors and opioid peptides, is implicated in the process for reinforcement and reward in the brain and nervous system by interaction with specific receptors [80]. The *opioid receptor mu(μ) 1* (*OPRM1*) gene encodes the μ-opioid receptor which is the principal target of endogenous opioid peptides and opioid analgesic agents such as beta-endorphin and enkephalins, which are associated with neurotransmission and pain modulation [81]. Positive reinforcement is mediated by both indirect (nicotine,

cannabinoids, alcohol) and direct (morphine) activation of the µ-opioid receptor [82]. Mechling et al [83] demonstrated that the µ-opioid receptor shapes the reward/aversion circuitry in the brain of rats. In humans, G-allele carriers at an A/G SNP in *OPRM1* have a dopamine-mediated lowered response to reward during positive reinforcement learning [84]. As a result of the link between reward and reinforcement, the *OPRM1* gene is associated with addiction in humans [85] whilst a different polymorphism (C77G) is linked to increased levels of alcohol consumption in male macaques [86]. This polymorphism is also associated with increased aggressive-threat behaviours in rhesus macaques [87] with G-allele carriers also showing higher baseline attachment behaviours than CC homozygous individuals [88]. Here, the relationship between the C77G polymorphism (rs195917455) and anxiety in rhesus macaques was investigated.

## Attention bias

AB describes a tendency to preferentially attend to emotional compared to neutral cues and is influenced by underlying affect [89, 90]. AB is a new approach to assessing animal cognition, affect and anxiety that has shown significant reproducibility between measures in rhesus macaques [91, 92]. Previous studies have shown that genotype can affect social attention in humans [93, 94]. However, there is no published evidence for an association between AB and genotype in rhesus macaques and the extent to which genetic variation influences attention to potentially threatening stimuli is not yet fully understood.

## Behaviour

Stressful events can have a significant effect on behaviour [95]. When faced with a potential threat, survival may depend on an appropriate behavioural change associated with either the active fight-flight or passive freeze-hide responses [96]. This behavioural change is often the most overt, easily observable, and biologically economical response to stress as a threat may be avoided by removing oneself from the stimulation, for example, an animal pursued by a predator will avoid the danger by escaping and a heat-stressed animal may move to find water or shade [97]. However, actively avoiding a stressor may not be possible for captive animals as this environment is associated with a range of uncontrollable factors from which the animal cannot escape, for example, inappropriate artificial lighting schedules, temperature variation away from their preferred ranges, space limitations and, sounds and smells of predator species [98]. Captive animals have limited or no opportunities for changing their spatial proximity to conspecifics and caretakers, environmental conditions, and nutrition.

Captive animals will have a different behavioural repertoire to their free-ranging counterparts [99]. However, free-ranging behaviours should be considered as it provides a reference of species-typical behaviour [100]. Free-ranging macaques spend most of their time allogrooming, resting, foraging, and exploring their environment [101–103]. These behaviours should apply to captive group or pair-housed macaques and a change in the magnitude or frequency of these key species-typical behaviours can be indicative of an underlying issue, such as ill health, pain, or stress [104]. In macaques, stress and anxiety can manifest as displacement behaviours and aggression [105]. Camus et al [105] reported five distinct behavioural profiles when recording spontaneous atypical behaviour in 40 captive cynomolgus macaques (*M. fascicularis*). One behavioural profile (E) was associated with a high occurrence of aggressive and displacement behaviours. Displacement behaviours are acts performed by the animal that are irrelevant to the behavioural context [106]. It is thought that these behaviours may occur as a form of energy dissipation or as a conflict between two competing motivations [107]. In primates, displacement behaviours include yawning, shaking, auto-grooming, and scratching

[108]. Facial expressions form a complex suite of behaviour associated with stress and communication. A lip smack is a characteristic affiliative or appeasement display, while a fear grimace, indicated by bared teeth, is a sign of fear or social submission [109]. Macaque facial movements have been described using a macaque Facial Action Coding System (MaqFACS) revealing similarity in facial expression across primate species [110]. However, several species-specific specialisations were also described with macaques having more independent control over their ear movements compared to humans or chimpanzees (*Pan troglodytes*) [110]. Ear movements are thought to play an important role in macaque social communication as grimace and lip-smack displays include prominent ear movements [111]. Enhanced vigilance is a further indicator of anxiety [108] with hypervigilance an indicator of extreme stress in many species [112]. The function of this behaviour is to detect social and predatory threats and, in response to stress, vigilance behaviour increases [113]. Socially stressed, subordinate macaques engage in vigilant scanning more frequently than dominant animals [114]. Vigilance is commonly accompanied by appeasement (lip smack) or fear (grimace) behaviours [115, 116]. Here, we examine variation at the nine previously described candidate loci and examine the role of variation at these loci in describing behaviour and AB in a group of captive rhesus macaques.

## Materials & methods

### Ethics

Protocols were developed following discussion with the facility Home Office Inspector (Nov 2011) and carried out in accordance with ethical guidelines for work with non-human primates [117]. Approval was granted by the Medical Research Council (MRC) Animal Welfare and Ethical Review Body (AWERB) in 2014, Roehampton University Ethics Committee (approval #LSC 14/113) and the Liverpool John Moores University (LJMU) ethics panel (approval #EB/2014-1). Further approval was granted by the MRC AWERB in November 2017, and the LJMU ethics panel (approval #EB_EH/2017-5). Animal health was monitored daily by the care staff, and annually with a full veterinary examination. No regulated procedures were carried out for this study and at no point were animals removed from their home colony at any time. Participation was voluntary insofar as only animals who approached the apparatus for rewards took part. Blood samples for DNA analysis were collected by a trained veterinarian with the primary purpose of colony management. Methods and results are reported according to the ARRIVE guidelines [118].

### Animals

Data were collected from 109 adult rhesus macaques (*Macaca mulatta;* 94 female) housed at the Centre for Macaques, MRC Harwell Institute (MRC-CFM), UK (mean age on first day of testing = 8.83 years, range = 2.5–18.3 years). The centre is licenced by the Home Office to breed macaques for provision to UK facilities. Rhesus macaques are a female philopatric species that have a linear hierarchy based on female relatedness [119]. To reflect their typical social system, monkeys at MRC-CFM are housed in social breeding groups comprising one adult male and breeding females. The monkeys that took part in the current research were housed in 29 breeding groups comprising one adult male and between three and 11 related females, plus infants and juveniles. Social rank for each female was categorised on a scale of high-mid-low, determined through consultation with facility staff who care for the animals on a daily basis, and from our published data [120]. All monkeys had previously been trained to station next to a target in order to take part in AB testing [120] and were familiar with researchers at the time testing was conducted. Each group had access to a free-roaming room

(3.35m × 8.04m × 2.8m) and an adjacent cage area (1.5m × 6.12m × 2.8m), accessible through hatches, with a minimum total space of 3.5m$^3$/breeding animal in the largest groups. Each free-roaming room had a large bay window at one end facing outdoors and allowing a natural day-night cycle. At the other end of each room was an internal window into the hallway used by staff. Internal windows were fitted with movable mirrors so that monkeys could manipulate the mirrors to view activities along the corridor. Rooms were furnished with wooden platforms and poles (horizontal, vertical, diagonal), fire hose, ladders, plastic horse jumps and saddle racks, PVC piping, plastic barrels and balls, and small plastic blocks attached to structures or walls. The floor was covered with a deep layer of straw and shavings. All rooms were temperature controlled (20 ˚C ± 5) with humidity at 55% ± 10%.

Animals were free to move between the room and cage area at all times during this research, and at no point were the hatches used to retain animals. Adjacent groups were able to see and hear each other from the cage area, but there was no possibility for physical contact. All testing took place in the cage area, which monkeys entered voluntarily, with open access to the free-roaming room at all times. The cage area had three tiers with open access points between them. Individuals tended to have preferred levels they would spend time in and so staff and researchers would work with an individual at their preferred location, a key factor when working with group housed animals.

The macaques were fed twice daily by scatter feed, morning and afternoon, with sufficient food to last for a 24 h period. The diet varied daily and included a dried forage mix (cereal, peas, beans, lentils etc.), a range of fruit and vegetables, bread and boiled eggs. Water was available *ad libitum* in both the room and cage area at all times. Further details, images and video of the facility can be found at: https://www.mrc.ac.uk/research/facilities-and-resources-for-researchers/mrc-centre-for-macaques/ and www.nc3rs.org.uk/macaques.

## DNA collection

Blood samples were collected by the Named Veterinary Surgeon at MRC-CFM during the macaques' annual health screening. Macaques were sedated with KHCl prior to this procedure. Blood samples, which were primarily used for colony management purposes, were collected into EDTA K2 (anticoagulant) tubes, centrifuged, and wrapped in cotton wool prior to transport to B&K Universal Laboratories (Marshall BioResources), Grimston, East Yorkshire, UK. DNA extraction occurred upon arrival at the laboratory using the DNeasy Blood & Tissue Kit from Qiagen (Catalogue no. 69506) and concentrations and purity measured using a Thermo Fisher Scientific Inc. NanoDrop™ Lite Spectrophotometer. The DNA samples were stored at -20˚C at Liverpool John Moores University, James Parsons Building, Liverpool, UK until required for analysis. Samples were stored at 4˚C during use.

## Genotyping

Eight genes were analysed for variation at a total of nine loci. For four loci the variants genotyped were length polymorphisms (*5-HTTLPR*, *STin*, *TPH2* and *MAOA*) whilst for the other five (*HTR2A*, *DRD4*, *OXTR*, *AVPR1a* and *OPRM1*) SNPs were analysed. Samples were genotyped for the four length variants using PCR amplification and gel-analysis. SNPs for *HTR2A*, *DRD4*, and *OXTR* were determined by amplification and sequencing, with haplotypes manually reconstructed from multi-locus genotype data, while SNPs for *AVPR1a* were determined through amplification followed by restriction enzyme digestion and gel-analysis [121]. Seven intron variants for *OXTR* (rs196783445, rs292502465, rs300857875, rs308701533, rs292035217, rs302789768, rs283226059) were identified using Ensembl [122] all within a short intron tractable for PCR and sequencing. A TaqMan genotyping assay was carried out

for the *OPRM1* C77G polymorphism [123]. PCR cycle times and temperatures were adapted from established protocols and optimised for the current study (Table 2). All PCR products were visualised using GelRed stain during electrophoresis; however, GelRed caused inconsistencies in the accurate sizing of the *5-HTTLPR* length polymorphism. Instead, ethidium bromide was used as the visualization dye for *5-HTTLPR*.

## Genetic data preparation

The final dataset consisted of genotypes from nine loci for each monkey (S1 Table). Variant information for each genotype and associated behaviours from the literature are shown in Table 2. For, *HTR2A*, *DRD4* and *OXTR*, the SNPs (3, 4 and 7 respectively) were manually phased into haplotypes prior to analysis. For *AVPR1a*, there were two upstream polymorphisms: a SNP (rs194455405) and an 8bp insertion (11:62913090) which were combined to create three haplotypes. For the genes for which sufficient published data were available (*5-HTTLPR*, *STin*, *TPH2*, *MAOA*, *OPRM1*) genotypes were reclassified prior to statical analysis as high or low expressing based on their known biological effects (high or low risk / expression; Table 3).

## Attention bias

A full AB protocol is presented in Howarth et al [92]. In brief, unfamiliar threat-neutral male conspecific face pair stimuli were shown to macaques for three seconds and their looking time to each stimulus pair recorded. Looking time towards face pair stimuli was blind coded from video [92]. Cognitive data were then averaged for baseline (assumed non-stressed state) and stress (following veterinary intervention) to create an AB profile for each animal.

## Behaviour

An ethogram of behavioural indices (Table 4) was constructed to include key behaviours related to aggressive, anxiety, distraction, fear, foraging, prosocial and inactive behaviour. These categories were chosen as they have previously been identified as key behavioural indices for wellbeing in rhesus macaques [105, 108, 109].

Continuous focal animal behavioural observations were completed using Behavioral Observation Research Interactive Software (BORIS) [129]. Observations were conducted following each AB trial with data collected on four consecutive weekdays (Tuesday—Friday) plus the following Monday. Following completion of the AB trials, groups were given 10 minutes to settle without observation before each monkey was observed for five minutes. Observations were conducted within 60 minutes of the cognitive trials.

## Cognitive data preparation

Cognitive data for social attention comprised two continuous response variables: time spent looking at the threat face per three second trial (THR; range = 0<3000ms) was included to assess interest in threat faces specifically, and total time spent looking at the threat-neutral face pair overall per trial (TL; range = 0<3000ms) was included to assess interest in social images regardless of emotion content.

To reduce the potential influence of outliers, following Howarth et al. [92], we removed cognitive trials for which a disruption was deemed to have occurred. Disruptions may be associated with increased stress, which our previous work has shown influences AB to threat [91]. Disruptions were identified as: receiving treatment for illness, a change to group membership

**Table 2. Primer pairs, location, and PCR conditions for amplification of *5-HTTLPR*, *STin*, *TPH2*, *HTR2A*, *MAOA*, DRD4, *OPRM1*, *OXTR*, and *AVPR1a*.** All genomic positions are from Mmul_1 [122, 124].

| Gene | Variant type | Primer and genomic location | PCR cycles & temperatures | Source |
|---|---|---|---|---|
| *5-HTTLPR* | Length | 5HTT-F<br>(5'-GCCGCTCTGAATACCAGCAC-3')<br>HTTLPR_intl<br>(5'-CAGGGGAGATCCTGGGAGGG-3')<br>Chromosome 16: 25,898,078–25,898,491 | 95˚C for 5 mins<br>40 cycles: 95˚C for 30 s, 61˚C for 30s, 72˚C for 1 min<br>Final extension: 72˚C for 7 mins | [125, 126] |
| *STin* | Length | STin-F<br>(5'-TGTTCCCAGACTTACACCAGTG-3')<br>STin-R<br>(5'-GTCAGTATCACAGGCTGCGAG-3')<br>Chromosome 16: 25,882,527–25,882,740 | 95˚C for 3 mins<br>35 cycles: 95˚C for 30s, 55˚C for 30s, 72˚C for 1 min<br>Final extension: 72˚C for 5 mins | Primers based on human primers from Ogilvie et al [39] redesigned based on *M. mulatta* sequence |
| *TPH2* | Length | TPH2-U3F5<br>(5'-TGTAGGAAACTTCTCATCACAA-3')<br>TPH2-U3R5<br>(5'-CAGCATAAAATTCATAGTCCCAAG-3')<br>Chromosome 11: 71,601,416–71,601,688 forward strand | 95˚C for 3 mins<br>35 cycles: 94˚C for 30s, 55˚C for 30s, 72˚C for 30s<br>Final extension: 72˚C for 5 mins | [43] |
| *HTR2A* | SNP | HTR2A-F<br>(5'-GGCATGACAAGGAAACCCAG-3')<br>HTR2A-R<br>(5'-CCAGGACATTTATCTCCCCGA-3')<br>Chromosome 17: 25,340,395–25,341,115 | 95˚C for 3 mins<br>35 cycles: 95˚C for 30s, 55˚C for 30s, 72˚C for 1 min<br>Final extension: 72˚C for 5 mins | Novel primers, no published PCR protocols |
| *MAOA* | Length | MAOA-F(2)<br>(5'-CAGAAACATGAGCACAAACG-3')<br>MAOA-R(2)<br>(5'-TACGAGGTGTCGTCCAAGTT-3')<br>Chromosome X: 43,576,337–43,576,618 | 95˚C for 5 mins,<br>40 cycles: 94˚C for 30s, 55˚C for 30s, 72˚C for 30s<br>Final extension: 72˚C for 10 mins | [127] |
| *DRD4* haplotype | SNP | DRD4-PROM-SNP-F<br>(5'-CGGGGGCTGAGCACCAGAGGCTGCT-3')<br>DRD4-PROM-SNP-R<br>(5'-GCATCGACGCCAGAGCCATCCTGCC-3')<br>Chromosome 14: 694,533–694,799 | 95˚ for 1min<br>35 cycles: 95˚ for 20s, 72˚C for 30s<br>final extension: 72˚C for 7 mins | [128] |
| *OXTR* | SNP | OXTR- F<br>(5'-CTGGACGCCTTTCTTCTTCG-3')<br>OXTR-R<br>(5'-AACTACTAGGGGCTTGGCTG-3')<br>Chromosome 2: 140,100,939–140,101,567 | 95˚C for 3 mins<br>30 cycles: 95˚C for 30s, 62˚C for 15s, 72˚C for 30s<br>Final extension: 72˚C for 5 mins | Novel primers, no published PCR protocols |
| *AVPR1a* | SNP | AVPR-F<br>(5'-AAGTCGGGAAGGTGAGCTC-3')<br>AVPR-R<br>(5'-CTTCCCGTAGCAAACACAGG-3')<br>Chromosome 11: 62,912,936–62,913,546 | 95˚C for 3 mins<br>35 cycles: 95˚C for 30s, 55˚C for 30s, 72˚C for 1 min<br>Final extension: 72˚C for 5 mins | Novel primers, no published PCR protocols |
| *OPRM1* | SNP | OPRM1_C77G_F<br>(5'- TGGCGCACTCAAGTTGCT-3')<br>OPRM1_C77G_R<br>(5'- GGGACAAGTTGACCCAGGAA-3')<br>Probes: OPRM1_C77G_VIC<br>(5'-CAGCACGCAGCCC-3') labelled with VIC for detecting G allele<br>OPRM1_C77G_FAM<br>(5'-CAGCACCCAGCCC-3') labelled with 6-FAM for detecting C allele<br>Chromosome 4: 60,308,969–60,309,025 | 95˚C for 15mins,<br>40 cycles: 92˚C for 15s, 60˚C for 1min | Newly designed Taqman assay |

**Table 3. Overview of variants for *5-HTTLPR*, *STin*, *TPH2*, *HTR2A*, *MAOA*, DRD4, *OPRM1*, *OXTR*, and *AVPR* with frequencies and risk / expression if known.**

| Gene | Allele_anc / allele_alt | Risk / expression | Associated behaviours from literature |
|---|---|---|---|
| *SLC6A4* 5-hydroxytryptamine transporter length-polymorphic repeat (*5-HTTLPR*) | L | High | Anxiety (active), Anxiety (low activity), Distraction, Fear, Prosocial |
| | S | Low | |
| *SLC6A4 STin* polymorphism | L | High | Anxiety (active), Anxiety (low activity), Distraction. |
| | S | Low | |
| *Tryptophan 5-hydroxylase 2 (TPH2)* | L | Low | Aggression, Anxiety (active), Anxiety (low activity). |
| | S | High | |
| 5-hydroxytryptamine receptor 2A (HTR2A) | C/G | No polygenic data | Anxiety (active), Distraction |
| | A/G | | |
| | A/C | | |
| *Monoamine oxidase A (MAOA)* | 55 | High | Aggression, Anxiety (active), Distraction. |
| | 56 | High | |
| | 57 | | |
| | 66 | High | |
| | 67 | | |
| | 77 | Low | |
| *Dopamine Receptor D4 (DRD4)* | A/T | No polygenic data | Aggression, Fear |
| | C/G | | |
| | C/G | | |
| | A/G | | |
| *Oxytocin receptor (OXTR)* | G/T | No polygenic data | Anxiety (active), Anxiety (low activity), Fear, Prosocial |
| | C/T | | |
| | C/G | | |
| | A/G | | |
| | C/T | | |
| | C/T | | |
| | A/G | | |
| *Arginine vasopressin receptor 1A (AVPR1a)* | T/C ± insertion | No polygenic data | Anxiety (active), Anxiety (low activity), Fear, Prosocial. |
| *Opioid receptor mu(μ) 1 (OPRM1)* | C/G | No polygenic data | Aggression, Anxiety (active), Anxiety (low activity), Distraction, Fear, Prosocial |

Abbreviations: **allele_anc**—ancestral allele; **allele_alt**—alternative allele

in the preceding week, injury in the preceding 48 hours, cleaning of housing in the last 24 hours and giving birth in the previous 24 hours.

## Statistical analysis

Statistical analyses were conducted in R v. 3.4.3v [130]. A maximal model was built per response variable separately. Response variables for social attention were THR, TL; Behaviour: aggression, anxiety, fear, distraction, prosocial. Each genetic variant was entered as a key predictor variable in each model. In addition, we controlled for factors known to influence socioemotional behaviour: sex, age, and condition [92]. All predictor variables were initially assessed to ensure none were correlated above 0.4 (which could result in collinearity [131]). Response variables were visually inspected for their distribution and transformed to obtain more normal distributions when necessary (this was the case for THR and TL). Appropriate transformations were identified using Tukey's Ladder of Transformation [132] to extract an appropriate

**Table 4. Ethogram of behaviours and groups for behavioural observation of captive rhesus macaques (*Macaca mulatta*).**

| Group | Behaviour | Description |
|---|---|---|
| **Aggressive** | Aggressive | The animal chases, attacks, threatens, stares at, displaces or lunges towards a conspecific. |
| **Anxiety (active)** | Body shake | Like a dog shake—the animal rapidly moves whole body, usually starting with shaking of the head followed by rest of body. |
| | Stereotypic | The behaviour has no obvious function. Includes pacing, bar biting or head tossing. |
| | Vigilance | The animal has alert posture scanning their environment or looking at a particular thing (may be out of view). |
| **Anxiety (low activity)** | Self-directed | The animal uses their hands or mouth to clean, scratch or manipulate their skin or fur. |
| | Sit hunched | The animal is sitting with their back and head curved round so that the head is below slumped shoulder. |
| | Yawn | Animal opens its mouth wide (not directed at conspecific) |
| **Distraction** | Object | The animal uses hands or mouth to investigate and move an inanimate, moveable object in the environment and / or pull or grab parts of the enclosure such as padlocks, sliding adjustable panels and cage dividers. |
| | Locomotion | Any behaviour (except those otherwise defined) that involves the animal moving from one location to another, for example, quadrupedal and bipedal walking and running, climbing, descending and jumping. The animal must not be engaged in any other activity. |
| **Fear** | Grimace | The animal's lips are pulled back to expose the teeth. |
| | Submissive | Animal moves away, flees, is displaced by or ducks away from a conspecific. They may give out high-pitched screams. Animal may also present its hindquarters in a non-sexual context (not followed by mating). |
| **Foraging** | Foraging | The animal is searching for and / or consuming food or water. |
| **Inactive** | Resting | The animal is lying horizontally with the stomach, back or side touching the floor or the animal us sitting upright |
| | Stationary | Weight baring on two or four legs. |
| **Prosocial** | Affiliative | Friendly interaction between the animal and a conspecific includes huddling, being in physical contact with a conspecific and hugging but not grooming behaviour. |
| | Allogrooming | Reciprocal grooming. The animal's skin or fur is cleaned, scratched or manipulated by the hands or mouth of a conspecific. The animal uses their hands or mouth to clean, scratch or manipulate the skin or fur of a conspecific. |
| | Interaction with baby | The animal interacts with a baby e.g., grooming, playing, carrying or feeding. |
| | Lip smack | Animal opens and closes its lips repeatedly without showing its teeth, occasionally making a smacking sound. |
| | Sexual behaviour | The animal presents or is presented the hindquarters. The animal is mounted or mounts. |
| **Other** | Other | Any behaviour not otherwise defined. |
| | Out of sight | The animal is not visible to the observer. |

lambda for transformation. Covariates were scaled using a z-transformation to a mean of zero and a standard deviation ± 1. Scaling continuous variables provides more comparable estimates for interpretation of model output [133]. Participant monkey identity was entered as a random factor in all models.

To account for relatedness between individuals, analyses were conducted using the animal model, in which pedigree was included as a random factor, using the MCMC-based Bayesian GLMM in R [134]. Ignoring family structure can increase type I error (false positive) rate

[135]. Bayesian analysis uses observed or "prior" distribution to estimate parameters of an underlying distribution [136]. For each model, an uninformative prior was specified following de Villemereuil [137]. An uninformative prior was used as the probability distribution was unknown and this type of prior allows for all distributions to be equally likely. Priors are determined based on observed data. As this is a new area of study, there is little prior knowledge that can inform the prior selection.

Each model was plotted and visually inspected to check for convergence. Parameters were set to obtain 1000 iterations, with a burn-in of 1000 and thinning interval of 500 to attain adequate plots, as assessed by eye. Independence of the data points was assessed by using a thinning interval producing autocorrelation of <0.1 assessed using the function autocorr [138]. With these parameters, absence of autocorrelation was assessed as <0.1 for animals at 10000 lag in all cases and the chain was said to have mixed well.

Meaningful effects were evaluated based on whether the 95% credible interval of the estimate overlapped zero [139]. We also provide P values in tables to aid readers in interpretation and for future meta-analyses. The use of Bayesian statistics with uninformative priors automatically controlled for multiplicity, therefore, it was not necessary to use a correction for multiple tests (e.g., Bonferroni correction as used in frequentist approaches [140]).

## Results

Genetic and cognitive data were collected from 109 monkeys (94 female). For 95 individuals (81 female), cognitive data were collected at both baseline and following a veterinary stressor. For 14 monkeys (13 female), cognitive data were collected at baseline only. Behavioural data were collected through observation of individuals in the home enclosure for 101 monkeys (93 female). For 77 individuals (69 female) behaviour data were collected at both baseline and following a veterinary stressor. For 12 females, data were collected at baseline only and for 12 females, following the stressor only.

At baseline, 796 trials were collected, and 154 trials were removed from the data set due to disruption, for example, unplanned veterinary visits, cleaning, or births. The cognitive data was then averaged for these trials (2–11 trials; mean = 6) leaving a total of 109 rows of baseline data. A total of 392 trials were collected following the stressor and no trials were removed from the data set. These were averaged for each monkey (1–5 trials; mean = 4) leaving a total of 95 rows of stress data.

### Novel haplotypes determined prior to analysis

SNPs for *HTR2A*, *DRD4*, *OXTR* and *AVPR1a* were determined by amplification and sequencing, with haplotypes manually reconstructed from multi-locus genotype data prior to analysis. These novel haplotypes, SNPs and dbSNP numbers are shown in Table 5. This is the first presentation of haplotypes for these polymorphisms in *HTR2A*, *DRD4*, *OXTR* and *AVPR1a*.

### Replication of previous evidence for an association of *5-HTTLPR* and *STin* with social attention and behaviour

Of the genes that were analysed with knowledge of which were potentially high and low expressing alleles, *5-HTTLPR* and *STin* were found to be significantly associated with measures of social attention and behaviour. For both *5-HTTLPR* and *STin*, LL was associated with high expression and the s-allele associated with low expression.

The duration of TL and the genotype for *STin* were significantly associated. Macaques with the low expressing alleles had a significantly greater duration of TL (SL, SS: 1.633 ± 0.605 s)

**Table 5. Sequences for each haplotype for *AVPR1a*, *HTR2A*, *DRD4* and *OXTR*.**

| Gene | Haplotype | Sequence |
|---|---|---|
| *HTR2A* | 1 | CAA |
| | 2 | GGA |
| | 3 | GGC |
| | 4 | GAC |
| | dbSNP rs labels: rs80365915, rs80363349, rs196407124 | |
| *DRD4* | 1 | AGGG |
| | 2 | TGGG |
| | 3 | ACGG |
| | 4 | TGCA |
| | dbSNP rs labels: rs300413141, rs1079355788, rs290724315, rs301203363 | |
| *OXTR* | 1 | TTGGCCG |
| | 2 | GCCATTA |
| | 3 | GTGGCCG |
| | dbSNP rs labels: rs196783445, rs292502465, rs300857875, rs308701533, rs292035217, rs302789768, rs283226059 | |
| *AVPR1a* | 1 | T; no insertion (-) |
| | 2 | C; no insertion (-) |
| | 3 | C; with insertion (+) |
| | dnSNP rs label: rs194455405 plus an 8bp insertion (11:62913090) which were combined to create three haplotypes | |

than macaques with the high expressing allele (LL; 1.401 ± 0.555 s, 95% CI [-19.049, -1.677]; Table 6).

The duration of distraction behaviour was associated with *5-HTTLPR* and, possibly, *STin*. Macaques with the low expressing alleles (SL, SS) for either of these polymorphisms performed more distraction behaviour (*5-HTTLPR*: 23.391 ± 27.239 s, *STin*: 25.749 ± 26.370 s) than macaques with the high expressing allele (LL) for *5-HTTLPR* (20.071 ± 18.296 s, 95% CI [1.466, 14.878]) or *STin* (14.918 ± 13.574 s, 95% CI [-14.260, 0.306]; Table 6).

Macaques with the high expressing allele for *5-HTTLPR* (LL) exhibited more fear behaviour (0.898 ± 1.250 s) than macaques with the low expressing alleles (SL, SS; 0.308 ± 0.769, 95% CI [1.603, 9.803]; Table 6). *TPH2*, *MAOA* and *OPRM1* were not associated with social attention or behaviour.

**Table 6. Relationship between high and low expressing alleles for *5-HTTLPR* and *STin* with the mean duration of social attention (n = 109; total duration looking at the threat and neutral faces (TL) and behaviour (n = 90) in rhesus macaques (*Macaca mulatta*)).** Credible intervals not crossing 0 are highlighted in bold.

| | | HE mean (s) | LE mean (s) | HE df | LE df | Lower CI | Upper CI | P |
|---|---|---|---|---|---|---|---|---|
| **TL** | *STin* | 1.401 ± 0.555 | 1.633 ± 0.605 | 45 | 62 | **-19.049** | **-1.677** | 0.030 |
| **Distraction** | *HTTLPR* | 20.071 ± 18.296 | 23.391 ± 27.239 | 46 | 42 | **1.466** | **14.878** | 0.022 |
| | *STin* | 14.918 ± 13.574 | 25.749 ± 26.370 | 33 | 55 | -14.260 | 0.306 | 0.034 |
| **Fear** | *HTTLPR* | 0.898 ± 1.250 | 0.308 ± 0.769 | 46 | 42 | **1.603** | **9.803** | 0.002 |

Abbreviations: **HE**–high expressing; **LE**–low expressing; **CI**–credible interval

## New evidence for an association of *DRD4* and *OXTR* with social attention

The durations of the measures of social attention were significantly associated with *DRD4* haplotype with individuals homozygous for haplotype 1 (1–1) associated with a short duration of THR and TL (THR: 0.922 ± 0.496 s, TL: 1.598 ± 0.788 s) compared to individuals heterozygous for haplotypes 2–3 (THR: 1.217 ± 0.470 s, 95% CI [0.323, 8.44]; TL: 1.930 ± 0.390 s, 95% CI [0.718, 41.402]) and haplotypes 2–4 (THR: 1.362 ± 0.746 s, 95% CI [2.001, 11.565]; TL: 2.068 ± 0.645 s, 95% CI [4.690, 54.698]; Table 7). The duration of TL was also associated with *OXTR* haplotype. Individuals heterozygous for haplotypes 1–3 had a significantly longer duration of TL (1.015 ± 0.836) compared to 2–3 individuals (0.766 ± 0.515 s, 95% CI [0.569, 39.556]). None of the polymorphisms for *HTR2A* or *AVPR1a* were associated with differences in the duration of THR or TL.

**Table 7. Relationship between the new polymorphisms with no previous published data expression or behaviour data (*HTR2A*, *DRD4*, *OXTR*, *AVPR1a*) and the mean duration of social attention (n = 109; duration looking at the threat face (THR), and total duration looking at the threat and neutral faces (TL)) and behaviour (n = 90) in rhesus macaques (*Macaca mulatta*).** Credible intervals not crossing 0 are highlighted in bold. Reference variant with most negative test value and a sample size greater than 8 were selected to aid interpretation for each gene.

| | Gene (ref variant) | Haplotype | Post mean | n | Lower CI | Upper CI | P |
|---|---|---|---|---|---|---|---|
| **THR** | *DRD4* (1–1; n = 27) | 1–2 | 0.631 | 42 | -1.198 | 2.638 | 0.498 |
| | | 1–3 | 2.020 | 3 | -2.5090 | 6.985 | 0.398 |
| | | 1–4 | 0.738 | 3 | -4.041 | 5.573 | 0.746 |
| | | 2–2 | 0.344 | 24 | -2.141 | 2.434 | 0.748 |
| | | 2–3 | 4.400 | 5 | **0.323** | **8.444** | 0.030 |
| | | 2–4 | 6.882 | 3 | **2.001** | **11.565** | 0.008 |
| **TL** | *DRD4* (1–1; n = 27) | 1–2 | 5.590 | 42 | -4.222 | 14.952 | 0.240 |
| | | 1–3 | 14.356 | 3 | -8.756 | 39.908 | 0.230 |
| | | 1–4 | -3.479 | 3 | -27.101 | 19.424 | 0.778 |
| | | 2–2 | 2.890 | 24 | -9.164 | 14.334 | 0.604 |
| | | 2–3 | 22.173 | 5 | **0.718** | **41.402** | 0.046 |
| | | 2–4 | 29.504 | 3 | **4.690** | **54.698** | 0.022 |
| | *OXTR* (2–3; n = 10) | 1–1 | 6.573 | 14 | -10.371 | 26.025 | 0.492 |
| | | 1–2 | 0.905 | 51 | -15.824 | 15.794 | 0.892 |
| | | 1–3 | 21.298 | 9 | **0.569** | **39.556** | 0.020 |
| | | 2–2 | 3.356 | 25 | -13.657 | 20.432 | 0.688 |
| **Anxiety (active)** | *AVPR1a* (BC; n = 14) | AB | 39.857 | 21 | -80.306 | 163.781 | 0.514 |
| | | AC | 278.582 | 2 | **24.414** | **514.356** | 0.018 |
| | | BB | 46.596 | 52 | -52.252 | 149.498 | 0.388 |
| **Fear** | *HTR2A* (1.1; n = 25) | 1.2 | -0.864 | 33 | -6.34 | 3.452 | 0.730 |
| | | 1.4 | -3.062 | 6 | -11.812 | 5.503 | 0.482 |
| | | 2.2 | 6.154 | 14 | **0.345** | **12.539** | 0.048 |
| | | 2.4 | 6.600 | 8 | -1.317 | 14.853 | 0.110 |
| | | 3.4 | 13.606 | 2 | **0.495** | **26.851** | 0.042 |
| | *OXTR* (2.3; n = 8) | 1.1 | 9.455 | 10 | **1.058** | **19.083** | 0.036 |
| | | 1.2 | 6.566 | 43 | -0.672 | 13.884 | 0.074 |
| | | 1.3 | 9.7805 | 7 | 0.8113 | 19.7451 | 0.052 |
| | | 2.2 | 8.4165 | 22 | -8.0152 | 0.8785 | 0.832 |

### New evidence for an association of *HTR2A*, *OXTR* and *AVPR1a* with fearful and anxious behaviour

The duration of fear behaviour was significantly associated with *HTR2A* haplotype. 1–1 individuals performed fear behaviour for a significantly shorter duration (0.394 ± 1.015 s) compared to individuals homozygous for haplotype 2 (0.751 ± 1.069 s, 95% CI [0.345, 12.539]) and heterozygous for haplotypes 3–4 (1.968 ± 2.178 s, 95% CI [0.495, 26.851]).

The duration of fear behaviour was associated with the *OXTR* haplotype. Individuals homozygous for haplotype 1 performed a significantly greater duration of fear behaviour (0.920 ± 1.127 s) than individuals heterozygous for haplotypes 2–3 (0.168 ± 0.315 s, 95% CI [1.058, 19.083]). 1–3 individuals also performed a greater duration of fear behaviour (1.335 ± 1.514 s) than 2–3 individuals and this difference approached significance (0.168 ± 0.315 s, 95% CI [0.811, 19.745]).

The duration of active anxiety behaviour was associated with *AVPR1a* genotype. AC individuals spent significantly longer performing active anxiety behaviour (165.091 ± 77.624 s) compared to BC individuals (68.003 ± 71.557 s, 95% CI [24.414, 514.356]). None of the haplotypes were significantly associated with aggressive, low activity anxiety, distraction, prosocial behaviour, or inactive behaviour.

## Discussion

We assessed the relationship between eight genes (nine loci) involved in emotion with two measures of social attention (the total time spent looking at the threat-neutral face pair overall per trial (TL) and the time spent looking at the threat face per three second trial (THR)), and eight behavioural categories in a relatively large sample of 109 rhesus macaques. Our results support previous findings that the 5-HTT length-polymorphic repeat in *SLC6A4* (*5-HTTLPR*) and intron 2 of the *serotonin transporter* (*STin*) are associated with distraction and fear behaviors. Additionally, we demonstrate for the first time that *STin* is also associated with social attention. We found an association between four novel polymorphisms in *5-hydroxytryptamine receptor 2A* (*HTR2A*), *Dopamine Receptor D4* (*DRD4*), *Oxytocin receptor* (*OXTR*) and *Arginine vasopressin receptor 1A* (*AVPR1a*) with measures of social attention and behaviour which we recommend for further investigation.

### Social attention

Three polymorphisms in candidate genes were found to be associated with measures of social attention. TL was associated with the genotype for the 16–17 bp VNTR polymorphism in *STin* and the haplotypes for both *DRD4* and *OXTR*. THR was also associated with the haplotype for *DRD4*. The influence of these variants on social attention and behaviour has not previously been studied in primates and, only the *STin* polymorphism has been explored in humans. No studies have explicitly looked at the link between attention bias (AB) and *STin*; however, the results of the present study and the available human literature suggest that the high expressing *STin* allele may be associated with increased anxiety and attention to social stimuli in both macaques and humans.

In macaques, the high expressing allele for *STin* (LL) was associated with a significantly shorter duration of TL than macaques with the low expressing allele (SL), i.e., those with the high expressing allele spent less time looking at social stimuli. This may suggest that individuals with the high expressing allele could be more susceptible to stress than individuals with the low expressing allele as previous cognitive studies have demonstrated that macaques become more avoidant of negative stimuli following a stressor [91].

Macaques and humans have different patterns of response to threat in cognitive trials using face stimuli: macaques showi greater avoidance of threat while anxious or stressed, compared to humans [141]. In humans, the high expressing *STin* allele (*STin*2.12) is associated with emotional disorders including anxiety, OCD, and depression [34, 35, 142]. OCD shares the underlying anxiety psychopathology with other disorders such as post-traumatic stress disorder and depression [143]. Cognitive factors implicated in the maintenance and development of OCD include the overestimation of threat and AB towards threatening information [144, 145]. Human participants with OCD struggle to disengage with disgust faces during cognitive trails of social attention [146, 147]. When shown threatening social stimuli, OCD patients spend significantly longer looking at the faces compared to healthy controls [147]. OCD is also associated with lower AB variability [148] and individuals with OCD showed lower within-individual variability of cognitive measure of social attention [149].

The polymorphisms for *DRD4* and *OXTR* presented here have not previously been studied in macaques or humans. Here, we found an influence of haplotype for both polymorphisms on social attention I macaques. *DRD4* individuals homozygous for haplotype 1 (1–1), were more avoidant of social stimuli compared to individuals heterozygous for haplotypes 2–3 or 2–4. For *OXTR*, individuals heterozygous for haplotypes 2–3 were more avoidant of social stimuli than those heterozygous for haplotypes 1–3. This suggests that macaques with *DRD4* haplotype 1 or *OXTR* haplotypes 2–3 may be more vulnerable to stress than macaques with other haplotypes for these genes. Other polymorphisms within these genes have been associated with a range of conditions (e.g., *DRD4*: anger, aggression, and delinquency [64, 65]; *OXTR*: reduced positivity [70], and poor social recognition skills [71]) and the results demonstrated here suggest that these novel polymorphisms could be key for future candidate gene studies for cognitive measures of social attention in rhesus macaques.

Further research to increase the sample size, and therefore the statistical power [150, 151] would be beneficial, especially for confirming the effects of the rarer haplotypes. However, the data presented here were from 109 macaques and the sample size was an improvement on other published rhesus macaque studies (e.g., n = 20 [152], n = 20 [153], n = 9 [154], n = 7 [155], n = 29 [28]).

## Behaviour

Six polymorphisms in candidate genes were found to be associated with differences in behaviour. Genotypes for both polymorphisms within the serotonin transporter (*5-HTTLPR* and *STin*) were associated with distraction behaviour. The duration of fear behaviour was associated with *5-HTTLPR* genotype and haplotype for *HTR2A* and *OXTR*. *AVPR1a* haplotype was associated with the duration of active anxiety behaviour. Apart from *5-HTTLPR* and *STin*, these variants have not previously been studied in macaques or humans. Our data suggest they are worth further investigation.

The low expressing alleles for both *5-HTTLPR* and *STin* were associated with a longer duration of distraction behaviour. S-allele carriers spent longer engaged in distraction behaviour compared to l-allele homozygotes for both *5-HTTLPR* and *STin*. In both humans and macaques, the s-allele for *5-HTTLPR* has a lower rate of serotonin transporter transcriptional efficiency [21, 22, 24]. Juvenile rhesus macaques homozygous for the s-allele displayed more anxious and threatened behaviour than l-allele carriers during novel fruit, human intruder, remote-controlled car, and free play tests [24].

Here, distraction behaviour included locomotion and manipulation of objects. Both behaviours have previously been identified as self-soothing responses to stress in primates. For example, juvenile macaques show an increase in locomotion following maternal separation

[156, 157], agitated locomotion increases following agonistic interactions with macaque conspecifics [158] and, orangutans (*Pongo pygmaeus*), frustrated during delayed computer trials, displayed more object manipulation behaviour than those not subject to the delayed trials [159]. However, it is important to note that both behaviours could be categorised as "exploratory" and classed as positive or neutral responses. For example, reduced locomotor activity could indicate withdrawal [160] and object manipulation is often considered a natural behaviour associated with tool use [161]. The difference in interpretation of locomotor and manipulation behaviours means that there is disagreement between the Bethea et al [24] study and the data presented here on the impact of *5-HTTLPR* genotype on affective state.

Bethea et al [24] reported that genotype for *5-HTTLPR* had a significant association with activity during a free play test. S-allele homozygotes were significantly less active than l-allele carriers. This reduced activity was measured using percentage of time engaged in locomotor behaviour and percentage of time spent playing with toys, the opposite of the results presented in the present study. Previous studies have also revealed that macaques with the lower expressing alleles have a heightened hormonal response to stress. In response to maternal separation, juvenile macaques heterozygous for the l-allele had lower adrenocorticotropic hormone (ACTH) levels than s/l heterozygotes [23, 153]. ACTH stimulates the adrenal cortices of the adrenal glands to release mineralocorticoid and glucocorticoid hormones, including cortisol [162] leading to higher cortisol responses to stress in s-allele homozygotes [28]. Fear behaviour is also influenced by *5-HTTLPR* genotype. In the present study, the high expressing allele (LL) for *5-HTTLPR* was associated with a longer duration of fear behaviour while Bethea et al [24] revealed s-allele homozygotes to engage in fear behaviour more frequently than l-allele carriers.

Much of the previous literature focuses on the interaction between *5-HTTLPR* genotype and response to a specific stressor, for example, separation [23, 153] or consecutive fear and anxiety eliciting tests [24]. Studies that applied no specific acute stressor showed no behavioural differences associated with genotype for *5-HTTLPR* [28]. In the present study, no specific fear or anxiety inducing tests were conducted. This project piggybacked onto the macaques' routine annual veterinary health check to ensure further stress was not caused as a result of this research. Data were collected before and after the health check and the procedures involved (e.g., ketamine sedation) have been shown to acutely compromise macaque welfare [91, 163–165]. Here we found no effect of the veterinary intervention on behaviour or measures of social attention. Refinements in veterinary and scientific practice and techniques involved in the veterinary health check [166] may mean that they are no longer sufficiently stressful to compromise welfare. In addition, all macaques were familiar with the veterinary intervention; novelty is known to be stressful for primates [167], therefore, this reduced novelty is likely to also have decreased the negative impact on their welfare and dampened the behavioural response.

The age and rearing environment difference between the studies may also be a contributing factor. Much of the previously published macaque literature focused on infant and juvenile macaques ($\leq$ 6 months [23, 153]; $\leq$ 1 year [24]) that experienced early life stress (i.e., early weaning). The typical age at which juvenile rhesus macaques stop suckling and are nutritionally weaned is between 10 and 14 months old [168–170]. Early weaning of captive macaques is defined as removal from the mother before 12 months and can be a source of considerable early life stress [171]. Peer-reared juvenile s/l heterozygote macaques, separated from their mothers at birth, had significantly heightened hormonal responses to separation stress compared to mother reared macaques of any *5-HTTLPR* genotype [23]. Peer-reared juvenile s/l heterozygote macaques, weaned at 30 days, also had significantly lower levels of 5-hydroxyindoleacetic acid in the cerebrospinal fluid (CSF 5-HIAA) compared to peer reared l-

homozygotes [124]. In these studies, the mother reared infants were also weaned before the recommended age (six months [23], seven months [124]).

Mother abused infant macaques with the *5-HTTLPR* s/l genotype had heighted anxiety and hormonal response to stress compared to infants that did not experience abuse, while s-homozygotes showed more behavioural reactivity (resistance to handling, tantrums) than l-homozygotes [153]. In this study, l/l homozygotes appeared to be buffered from the physiological effects of mother abuse and had lower levels of cortisol than non-abused infants of the other genotypes. The authors suggested that l-homozygotes are better able to adapt to initially heightened cortisol levels caused by their mothers' abuse.

The macaques involved in the present study were housed at the MRC-CFM. MRC-CFM has been involved in several research projects focusing on weaning including Prescott et al [171] and as such macaques are not weaned before 10 to 14 months [172]. As a result, only a small number of the older macaques had been weaned early and only two were homozygous for the *5-HTTLPR* s-allele. Early weaning was not included in the analysis here due to low sample size resulting from improved management practices and, therefore, the impact of the interaction between early life stress and the low expressing genotype cannot be explored in this data set.

Fear behaviour was associated with haplotype for *HTR2A* and *OXTR*. *HTR2A* encodes the serotonin 2A receptor which plays an important role in the modulation of mood and anxiety [48, 173, 174]. Polymorphisms in *HTR2A* have been associated with neuropsychiatric disorders including impulsive behaviour and schizophrenia in humans [49, 50]. In mice (*Mus sp.*), *HTR2A*-expressing cells in the central amygdala control innate and learned freezing [175]. Freezing is a characteristic fear response in mice [176]. Inactivation of the *HTR2A*-expressing cells downregulated learned-freezing response and upregulated innate freezing response. In marmosets (*Callithrix jacchus*), mRNA expression levels of *HTR2A* were shown to have a selective-relationship between trait-like anxiety suggesting that the serotonin receptor may only have a limited role in regulating serotonin signaling in the downstream pathway [177]. The authors pointed to potential sample size issues (n = 13); highlighting why further studies on polymorphisms within *HTR2A* are important for determining the role of this gene within anxiety and fear behaviours.

*OXTR* encodes G-protein coupled receptor proteins for oxytocin. Oxytocin is a neuromodulatory hormone that plays a counter-regulatory role in stress and fear responses [67, 68]. Other polymorphisms within *OXTR* have been shown to be associated with differences in oxytocin receptor density in the brain and Depression Anxiety and Stress Scale (DASS) scores in humans [178]. No *OXTR* polymorphisms have previously been studied in macaques and therefore, the polymorphisms presented here represent important variants for future study.

*AVPR1a* haplotype was associated with the duration of active anxiety behaviour. The polymorphisms investigated here were RS3 alleles which, in humans, are associated with altruism [75], pair-bonding behaviour [76], social behaviour including sibling conflict [77] and reproductive behaviour [78]. In humans, the high-risk genotype is 327 bp and the low risk in non-327 [179] with the 327 bp allele being associated with lower altruism and non-optimal mothering [180]. Other alleles, for example, the 334 bp allele was associated with relationship and marital problems [79]. These human variants do not exactly map onto the macaque polymorphisms studied here but point to *AVPR1a* haplotype influencing general and social anxiety in primates.

## Conclusion & future applications

We analysed eight genes for variation at a total of nine loci with cognitive and behavioural correlates in rhesus macaques. Of these, five had not previously been studied in primates and four

had not previously been studied in humans. Our results revealed a significant association between *STin* genotype, *DRD4* haplotype and *OXTR* haplotype and cognitive measures of social attention in rhesus macaques. Genotype for *5-HTTLPR*, *STin* and *AVPR1a* and haplotype for *HTR2A* and *OXTR* were significantly associated with the duration of behaviours including fear and anxiety. Due to the novelty of these findings, we recommend future work focus particularly on the association between *HTR2A*, *DRD4*, *OXTR* and *AVPR1a* with cognitive and observational measures of behaviour associated with wellbeing and survival in both macaques other species including humans. Understanding the biological underpinnings of individual variation in negative emotion (e.g., fear and anxiety), together with their impact on social behaviour (e.g., social attention including vigilance for threat) has application for managing primate populations in the wild and captivity, and has translational value for understanding the evolution and function of negative emotion in our own species.

## Supporting information

**S1 Table. Final dataset of cognitive and behaviour data with genotypes from nine loci for each rhesus macaque (*Macaca mulatta*).**
(XLSX)

## Acknowledgments

We thank the staff at MRC-CFM, particularly Faye Peters and Sebastian Merritt who assisted with data collection.

## Author Contributions

**Conceptualization:** Craig S. Wilding, Emily J. Bethell.

**Formal analysis:** Emmeline R. I. Howarth, Craig S. Wilding, Emily J. Bethell.

**Funding acquisition:** Emily J. Bethell.

**Investigation:** Emmeline R. I. Howarth, Isabelle D. Szott.

**Resources:** Claire L. Witham.

**Software:** Claire L. Witham.

**Supervision:** Claire L. Witham, Craig S. Wilding, Emily J. Bethell.

**Writing – original draft:** Emmeline R. I. Howarth.

**Writing – review & editing:** Emmeline R. I. Howarth, Isabelle D. Szott, Claire L. Witham, Craig S. Wilding, Emily J. Bethell.

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
