## [Decision Letter · Decision Letter 0]

18 Apr 2023

PONE-D-23-06906Genetic polymorphisms in the serotonin, dopamine and opioid pathways influence social attention in rhesus macaques (Macaca mulatta)PLOS ONE

Dear Dr. Bethell,

Thank you for submitting your manuscript to PLOS ONE. After careful consideration, we feel that it has merit but does not fully meet PLOS ONE’s publication criteria as it currently stands. Therefore, we invite you to submit a revised version of the manuscript that addresses the points raised during the review process.

The manuscript is interesting as the report will be interested to many audience who are interested in understanding molecular mechanisms underlying social behavior. I would request the authors resubmit it after thoroughly revising considering the comments raised by the reviewers. I would also request the authors to insert some points on how the finding can reflect on humans with respect to those behaviors and what the future steps that can be taken up using your findings.   

We look forward to receiving your revised manuscript.

Kind regards,

Asem Surindro Singh, Ph.D

Academic Editor

PLOS ONE

Journal Requirements:

2. In order to comply with PLOS ONE's guidelines for non-human primate experiments (http://journals.plos.org/plosone/s/submission-guidelines#loc-non-human-primates), please provide additional details regarding housing conditions, feeding regimens, environmental enrichment, and all relevant steps taken to alleviate suffering (anesthesia, analgesia, details about humane endpoints, euthanasia, etc.). Also indicate how often animal care staff monitored the health and well-being of the animals and the criteria used to make such assessments. Lastly, specify the disposition of animals at the end of the study (e.g. euthanasia, returned to home colony, etc.). If animals were euthanized following the study, please provide the method of sacrifice.

"This research was supported by NC3Rs grant NC/L000539/1to EJB. ERIH was supported by an LJMU PhD studentship."

4. Please expand the acronym “NC3Rs and LJMU” (as indicated in your financial disclosure) so that it states the name of your funders in full.

Reviewers' comments:

Reviewer's Responses to Questions

**Comments to the Author**

1. Is the manuscript technically sound, and do the data support the conclusions?

Reviewer #1: No

Reviewer #2: Yes

2. Has the statistical analysis been performed appropriately and rigorously? 

Reviewer #1: No

Reviewer #2: Yes

3. Have the authors made all data underlying the findings in their manuscript fully available?

Reviewer #1: Yes

Reviewer #2: Yes

4. Is the manuscript presented in an intelligible fashion and written in standard English?

Reviewer #1: Yes

Reviewer #2: Yes

5. Review Comments to the Author

Reviewer #1: Major revision recommended, but I doubt that the authors can address my concerns. Although I've provided an opportunity to the authors to revise their paper based on my comments, I have little hope that the authors can properly address my statistical concerns, particularly because they are all non-statisticians.

Reviewer #2: Very well planned study, executed meticulously with all necessary information. I did not find any major shortcomings in this article. It can be straight way go for publication. My best wishes to the authors.

6. PLOS authors have the option to publish the peer review history of their article (what does this mean?). If published, this will include your full peer review and any attached files.

Reviewer #1: No

Reviewer #2: No

---

## [Author Response · Author response to Decision Letter 0]

1 Jun 2023

We thank the two anonymous Reviewers for their useful feedback on the manuscript. We hope we have addressed all comments adequately. Reviewer comments are addressed below in order. Review comments shown in bold.

Response to Reviewer 1.

We are pleased that Reviewer 1 concluded that ‘the article is well-written and contributes significantly to the existing literature on the effect of genetic polymorphism on the negative emotion and social behaviour of primates’. In addition, we would like to thank Reviewer 1 for their suggestions to improve the manuscript further. We trust we have addressed all the minor and major points thoroughly as follows:

The name of the procedure implemented in this article is Bayesian MCMCglmm and it can mislead the reader. The author should use some alternative standard name for their procedure, for example, MCMC-based Bayesian GLMM. Note that the R package which implements MCMC based Bayesian GLMM method indeed is named “MCMCglmm”. However, the method should not be referred to as MCMCglmm.

We have changed the text where this is referred to in two places. In the Abstract, the text now reads “Genetic data were then correlated with cognitive and observational measures of behaviour associated with wellbeing and survival using MCMC-based Bayesian GLMM in R to account for relatedness within the macaque population.” On page 21 (tracked changes version for all page numbers) we have changed the text to read “To account for relatedness between individuals, analyses were conducted using the animal model, in which pedigree was included as a random factor, using MCMC based Bayesian GLMM in R [134].”

The abbreviation NHP (line 229) is used without clarification. Please clarify all the abbreviations carefully and introduce them in the right place in the article. Also, use abbreviations wherever possible. Use abbreviation instead of non-human primates at line 249.

We have now deleted the use of NHP and changed this to simply primates (page 11) and checked through the manuscript for additional abbreviations which require clarification. We have made additional changes to clarify abbreviations for MRC and LJMU (page 12). We would like to highlight that abbreviations used in tables are included in the table footnotes. 

As the same data is used repeatedly to test the significance of genetic polymorphisms on different response variables using different models, so an adjusted p-value in place of an ordinary p-value should be reported. In that case, however, it seems that some of the significant discoveries would turn out to be false discoveries. This is a major concern and must be addressed carefully.

We thank the reviewer for their concerns around this point and have re-stated the results focusing on credible intervals (the Bayesian form of confidence intervals) as they are more appropriate than P values for reporting Bayesian analysis. We have added a statement on page 22 to explain how we assessed significance: ‘Meaningful effects were evaluated based on whether the 95% credible interval of the estimate overlapped zero [140]’. We have replaced reported P values with CIs throughout the results section, and retained P values in the tables as additional information.

We understand the point Reviewer 1 has raised about multiple testing. The purpose of this paper is to identify possible new polymorphisms of interest in the study of social attention. Therefore, as stated in the manuscript we have taken an exploratory approach, ensuring that we identify possible variants of interest. This is to help other researchers in identifying new genetic variants to focus future research efforts on. A more conservative approach would remove some of these variants from consideration. We trust the educated readership of PLOS ONE will understand this distinction and will be able to interpret the credible intervals in this light. We have clarified in the text on p22 that we provide P values as secondary information (e.g. for future meta analyses), and so it is not necessary to use adjusted P values as they are not used for significance testing. Regarding the use of separate models, these were informed by our previous work detailed in Howarth et al (2021). Therefore, the model parameters are prior justified.

The discussion section is too long. It can be shortened by discarding lines like “When treated with mirtazapine, a noradrenergic and specific serotonergic antidepressant, individuals with the STin 12 allele showed a significant reduction in their Hamilton Depression Rating Scale score compared to individuals with the low expressing STin allele. Mirtazapine has also been used in the treatment of social anxiety [149] and social phobia [150].”

We are attempting in this discussion not just to discuss these results as they pertain to macaque behaviour, but also with some reference to what is currently known about humans, and so the translational value of our findings. Thus, we do not believe that we can significantly shorten the discussion. However, the section reading “When treated with mirtazapine, a noradrenergic and specific serotonergic antidepressant, individuals with the STin 12 allele showed a significant reduction in their Hamilton Depression Rating Scale score compared to individuals with the low expressing STin allele. Mirtazapine has also been used in the treatment of social anxiety [149] and social phobia [150]” has been removed as the essence of this study is captured in the previous section. We have also made some edits to the previous paragraph on p30 as shown by track changes. 

Tuning MCMC is a challenging problem and improper selection of burn-in may lead to the incorrect posterior distribution, which in turn may incorrectly interpret a non-significant covariate to be significant. It would be better if the authors demonstrate some of the MCMC trace plots and posterior density as supplementary figures. 

We appreciate the opportunity to clarify our procedure here. As stated in the manuscript this is why we used an uninformative prior, and visually checked plots for effective burn-in and posterior density by eye. We have added clarification on p22: “With these parameters, absence of autocorrelation was assessed as below 0.1 for animal at 10000 lag in all cases and the chain was said to have mixed well”. We do not have access to the plots as they were inspected at the same time as running the models and not saved. The Pedantics R package has been archived meaning we cannot easily run this code again to re-generate the plots. 

That said I think the article is novel and after addressing the above-mentioned concerns it can be accepted for publication.

We thank the referee for this summary and their pertinent feedback points.

Response to Reviewer 2:

We are pleased that Reviewer 2 found the research to be ‘well planned, executed meticulously with all necessary information … it can be straight way go for publication’. We feel this Reviewer understood the purpose of our paper in providing information that we trust will be of use for other researchers in identifying genes of interest in future studies. In particular, we feel the new knowledge about the potential role of genetic variants, STin, DRD4 and OXTR, in social attention in rhesus macaques, will facilitate new studies in this area.

---

## [Editor Report · Decision Letter 1]

21 Jun 2023

Genetic polymorphisms in the serotonin, dopamine and opioid pathways influence social attention in rhesus macaques (Macaca mulatta)

PONE-D-23-06906R1

Dear Prof Al-Eitan,

We’re pleased to inform you that your manuscript has been judged scientifically suitable for publication and will be formally accepted for publication once it meets all outstanding technical requirements.

Kind regards,

Asem Surindro Singh, Ph.D

Academic Editor

PLOS ONE
---

## [Editor Report · Acceptance letter]

10 Jul 2023

PONE-D-23-06906R1 

Genetic polymorphisms in the serotonin, dopamine and opioid pathways influence social attention in rhesus macaques (*Macaca mulatta*). 

Dear Dr. Howarth:

I'm pleased to inform you that your manuscript has been deemed suitable for publication in PLOS ONE. Congratulations! Your manuscript is now with our production department. 

Kind regards, 

on behalf of

Dr. Asem Surindro Singh 

Academic Editor

PLOS ONE